# A Portal to Visualize Transcriptome Profiles in Mouse Models of Neurological Disorders

**DOI:** 10.3390/genes10100759

**Published:** 2019-09-26

**Authors:** Rami Al-Ouran, Ying-Wooi Wan, Carl Grant Mangleburg, Tom V. Lee, Katherine Allison, Joshua M. Shulman, Zhandong Liu

**Affiliations:** 1Department of Pediatrics, Baylor College of Medicine, Houston, TX 77030, USA; rami.al-ouran@bcm.edu; 2Jan and Dan Duncan Neurological Research Institute, Texas Children’s Hospital, Houston, TX 77030, USA; 3Department of Neurology, Baylor College of Medicine, Houston, TX 77030, USA; 4Department of Molecular and Human Genetics, Baylor College of Medicine, Houston, TX 77030, USA; 5Department of Neuroscience, Baylor College of Medicine, Houston, TX 77030, USA

**Keywords:** neurodegeneration, neurological disorders, mouse models, transcriptomics

## Abstract

Target nomination for drug development has been a major challenge in the path to finding a cure for several neurological disorders. Comprehensive transcriptome profiles have revealed brain gene expression changes associated with many neurological disorders, and the functional validation of these changes is a critical next step. Model organisms are a proven approach for the elucidation of disease mechanisms, including screening of gene candidates as therapeutic targets. Frequently, multiple models exist for a given disease, creating a challenge to select the optimal model for validation and functional follow-up. To help in nominating the best mouse models for studying neurological diseases, we developed a web portal to visualize mouse transcriptomic data related to neurological disorders. Users can examine gene expression changes across mouse model studies to help select the optimal mouse model for further investigation. The portal provides access to mouse studies related to Alzheimer’s diseases (AD), Parkinson’s disease (PD), Huntington’s disease (HD), Amyotrophic Lateral Sclerosis (ALS), Spinocerebellar ataxia (SCA), and models related to aging.

## 1. Introduction

The use of model organisms is an essential step in the path to understanding neurological diseases and their progression. While several mouse models exist for studying neurological disorders, nominating the best model for functional follow-up is not an easy task. With sequencing technology becoming common and affordable, resources with a large collection of high-throughput transcriptomic profiles have been created, and data are made accessible through centralized portals. There are portals with large pools of samples from single nation-wide projects such as the Genotype-Tissue Expression (GTEx) project [1] with pools of non-disease tissues from human samples and The Cancer Genome Atlas (TCGA) (https://www.cancer.gov/tcga) with tumors/adjacent normal tissues from tumor patients. These portals allow users to download the processed expression data for further analysis. Alternatively, there are portals of data from studies focusing on particular tissues, such as the Allen Brain Atlas (https://portal.brain-map.org/) with brain tissue data and the Brain RNA-seq portal [2,3] with brain cells expression. These two databases have expressions from both human and mouse experiments and allow users to search and compare the expression changes of genes between the human and mouse experiments. This feature is very valuable as insight gained from model organisms is essential to understanding diseases and their progression in human. In addition, there are numerous additional studies from different laboratories which profiled model transcriptomes and deposited the data into databases such as the Gene Expression Omnibus (GEO) database. However, the collection from each of these studies is relatively small in size, and the data is available in various file formats, ranging from raw files, such as sequencing reads to normalized expression or analysis results from an array of methods. There are few centralized portals with collections of expression data deposited in GEO, such as the Gene eXpression Database (GXD) [4] which allows users to obtain the expression of single or multiple genes from mouse experiments. However, it does not provide comparisons to human expression data. Another portal, the ARCHS^4^ [5], has human and mouse studies from the GEO database processed through a unified pipeline and gene counts available for download for further processing and analyses. ARCHS^4^ allows users to search for studies using meta-data such as tissues and cell lines, a GEO study identification number, or search using gene symbols which shows tissue expression levels and finds genes with similar co-expression patterns. However, it is difficult to gain specific insights on genes associated with a certain disease as the identification for relevant studies from the vast space of all studies included in ARCHS^4^ is challenging. Furthermore, it does not allow gene expression changes investigation in the context of a specific experiment. To date, there is no centralized platform available where the mouse models could be jointly and fairly compared or evaluated using a set of target genes to comprehensively and specifically study neurological diseases in humans.

In a recent study [6], a large collection of mouse studies related to neurological disorders were collected and processed using a unified RNA-seq pipeline utilizing Amazon Web Services (AWS). The studies were collected from the GEO database and the Accelerating Medicines Partnership-Alzheimer’s Disease (AMP-AD) Knowledge portal and include nine neurological diseases, 14 cell types, and 251 mouse experiment-control comparisons (i.e., Differentially Expressed Genes (DEGs)). The analysis and processed data are available at https://www.synapse.org/#!Synapse:syn16779040, but there is no efficient way to examine genes of interest across all the processed studies without downloading the large dataset. Since this represents a rich resource of data which could be utilized by the research community, there is a need for a portal to access this data efficiently and intuitively. To address this shortcoming, we developed a centralized portal available at (http://mmad.nrihub.org) to access and visualize the mouse transcriptomic data from a comprehensive set of studies on neurological disorders processed by Wan et al. [6].

Using the portal, users can efficiently look up their genes of interest in a large number of mouse studies related to neurological disorders and draw insights about their genes of interest, which could help identify the best mouse model for their planned experiments or functional follow-up. The portal could be queried with a set of gene symbols and produces an interactive heatmap which displays the expression changes of queried genes across all included mouse data sets (e.g., experiment-control comparisons or DEGs). Our portal provides several advantages over existing resources for user query-based lookup of gene expression data in experimental data. First, it accumulates mouse model studies related to neurological disorders into a centralized portal with annotations added to each study for efficient searching. Second, the portal is specifically focused on mouse models related to neurological disorders. 

## 2. Materials and Methods 

### 2.1. Infrastructure

The transcriptome portal was constructed from three major parts, as shown in Figure 1. The “Web Portal” is the web interface which users directly interact with. In addition to basic features similar to other web applications, such as the search function, we included customization and interactive graph features to allow users to easily navigate and interpret data from hundreds of transcriptomes and tailor the output to suit their objectives. Moreover, customized graphs could be generated for reporting or publication purposes through the save function. A detailed description of all the functions implemented will be discussed in Section 2.2. 

“Data Sources” denotes the source from where the data was obtained. The “Application Development” component is the central part linking the “Web Portal” and “Data Sources”. It consists of all the coding scripts which control the transcriptome portal. Based on their roles, “Application Development” scripts could be divided into two categories: (1) user-interface code that implements the plots, tabs, input box, and other functions of the web interface; (2) processing code which handles the server settings, data processing, such as extract-transfer-load (ETL) pipeline, and post data processing, such as filtering and data checking. The scripts interact within and between the two categories in real-time while users access the portal.

### 2.2. Web Interface 

We developed a web interface, as shown in Figure 2. The user starts by uploading a file of gene names (text file with one gene symbol per line) or copying and pasting a list of gene symbols into a text box followed by choosing the required options, such as the minimum fold-change of genes, false discovery rate (FDR), clustering method, and desired annotation. The portal provides three tabs as output. The first tab displays a heatmap of the log fold-change of the experiment-control comparisons or DEGs of the input genes across the mouse model studies. Hierarchical clustering on both rows and columns is available as an option while generating the heatmap, and dendrograms are displayed on the top and right of the heatmap to provide an overview of the clustering structure of the mouse models and genes respectively. Users can use these dendrograms as a guide to identify clusters with similar expression changes. The heatmap is interactive such that users can zoom in to focus on a select set of genes and studies, and the heatmap could be downloaded as an image or a tab-delimited text file. Additionally, there are annotations for the mouse studies including disease, cell type, and sex at the top of the heatmap which can help users further comprehend and contextualize the patterns of gene changes across mouse studies. Users could also choose to view only subsets of mouse studies or group the studies by a certain disease, sex, or cell type. The second tab displays a table of the study identifiers (IDs) from the heatmap with their detailed annotations. The third tab displays interactive volcano plots for up to three studies at a time for a more detailed assessment of gene expression changes across mouse models. Users can highlight genes of interest on the volcano plots and can download the plots as images as well. Below each volcano plot, the complete list of genes per study with their log fold-change and FDR values are displayed in a table.

The web interface was developed using Shiny v.1.2.0 [7] and R v.3.5.2. The interactive heatmap was generated using the R package heatmaply v.0.15.2 [8].

## 3. Results and Discussion

In order to illustrate the use of our portal, we study three different “use cases”, including gene sets based on human transcriptional and genetic data. We examined two distinct Alzheimer’s disease (AD)-associated transcriptional signatures, based on published coexpression analyses from human postmortem brains [9,10]. First, in an analysis of 1647 samples from three brain regions (dorsolateral prefrontal cortex, visual cortex, and cerebellum), Zhang et al. [9] identified a promising module (yellow), consisting of 1098 genes. Using an integrative, network-based approach, the yellow module was ranked highest for its association to late-onset AD and was highly enriched with immune and microglial functions. We utilized our transcriptome portal to examine whether genes from the yellow module are differentially expressed in mouse models relevant to AD and many other neurological disorders.

Figure 3a shows the heatmap produced using the 1098 genes from the yellow module. From the heatmap, we could identify three regions of interest with high or low log fold-change values (R1, R2, R3). R1 consists of 14 DEGs (e.g., mouse comparisons), 12 of the DEGs are derived from seven AD-related studies, with predominately upregulated genes (Figure 3b). On the other hand, R2 (six DEGs) shows a set of significantly downregulated genes in models for studying different cell types in the mouse (Figure 3c). Lastly, R3 consists of five DEGs with genes highly upregulated, and the mouse DEGs (comparisons) are from three studies related to inflammation and immunity (Figure 3d). This is consistent with findings from human brain samples observed in the original study [9]. The inspection of individual subsets from the vast collection of mouse model gene expression data allows users to deeply interpret the results with ease. The ease of interpretation is facilitated by the interactive heatmap provided through the portal, which allows users to zoom into specific regions of interest and the option to download the heatmap. 

In a second example, we explored another co-expression module (m109) which was significantly associated with cognitive decline based on a more recent study by Mostafavi et al. [10]. The authors studied RNA-seq data from the human Dorsal Lateral Prefrontal Cortex (DLPFC) brain region in two cohorts: the Religious Orders Study (ROS) and the Rush Memory and Aging Project (MAP), collectively referred to as (ROSMAP), which consisted of 478 subjects. Sets of co-expressed genes (modules) were identified, and association analysis was performed between the co-expression modules and clinical traits. Using this approach, the authors found that m109 had a significant module-level association with cognitive decline. The top 112 reported genes with high levels of association with cognitive decline from m109 were used to examine their expression changes across mouse model studies. 

Figure 4a shows the heatmap produced using the 112 genes from m109. There are two regions with consistent expression changes across the majority of genes: R1 (four DEGs) with genes consistently upregulated and R2 (four DEGs) with genes consistently downregulated. In region R1 (Figure 4b), three DEGs (M132, M133, M140) are from the same paper which studied the effect of ethanol on synaptic transcriptome and synaptic plasticity [GSE73018], while M52 is from a paper investigating the biology of microglia [11]. In region R2 (Figure 4c), two DEGs (M48, M63) are from a paper investigating the transcriptomic profiles of different cell populations in the mouse brain, with M48 on the neurons and M63 on dopaminergic neurons from the midbrain [12]. The other two DEGs (M136,M138) are from the same study as (M132,M133,M140) shown in R1. This is interesting because it implies that input genes’ expression levels are significantly altered in opposite directions depending on which experimental treatment was used (M136,138 vs. M132,133,140). Faced with such a result, users might investigate the original study so that they may better understand its experimental design to determine the causal mechanism.

Lastly, we also used our tool to interrogate a set of the 90 susceptibility gene candidates for Parkinson’s disease (PD) based on a recently reported Genome-wide Association Study (GWAS) [13]. Figure 5a shows the resulting heatmap, and Figure 5b shows the sets of DEGs (M49, M46, M52) with high log fold-change which study three cell types in mouse (newly formed oligodendrocyte, myelinating oligodendrocyte, and microglia) respectively. Figure 5c shows a set of DEGs (M211, M214, M213, M212) all from the same study [14] of microglial neurodevelopment.

Using our portal, we can efficiently examine gene expression changes across a large number of mouse studies related to neurological disorders which can help in the identification of the most appropriate mouse model for further study and functional validation. It is of importance to mention that while we do nominate mouse models for functional follow-up, we do not provide a significant measure or ranking score to the nominated models, thus we recommend to closely examine the nominated mouse model study to first check how well does a nominated model fit the users’ hypothesis and study goals and how closely does it follow the transcriptional changes in the human data the user is trying to validate, and second to check if the nominated mouse model does represent or model the human condition the user is investigating.

## 4. Conclusions

In this study, we developed a portal to explore the transcriptome of mouse studies related to neurological disorders, and we created a web portal to access the studies. Our goal is to provide a resource where researchers can explore expression changes in their genes of interest across different mouse models related to neurological disorders. We anticipate that this functionality will facilitate nomination of the best mouse model for validation and functional lookup. Finally, additional types of analyses and visualizations to aid in the network-based analysis of AD across multiple species will be provided in the future, and we plan to keep adding additional mouse studies related to neurological disorders.

## Figures and Tables

**Figure 1 genes-10-00759-f001:**
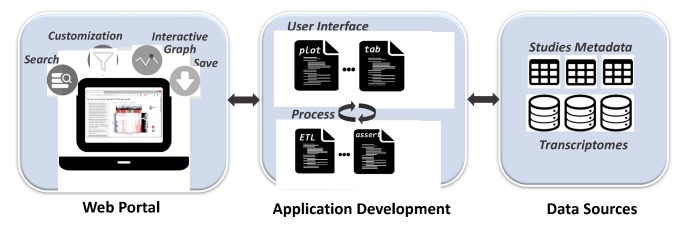
Infrastructure of the mouse transcriptome portal.

**Figure 2 genes-10-00759-f002:**
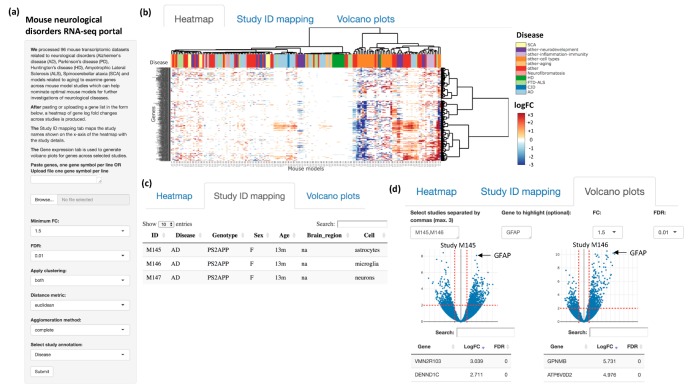
Web interface. (**a**) Users can input a list of genes into the text box or upload a file with a list of genes to study gene expression changes across mouse studies. (**b**) The ‘Heatmap’ tab displays an interactive heatmap with annotations at the top (e.g., disease). (**c**) The ‘Study identifier (ID) mapping’ tab displays a table mapping between the mouse model IDs displayed on the heatmap and the full study annotations. (**d**) The ‘Volcano plots’ tab displays interactive volcano plots where users could compare three studies at a time and highlight a gene of interest. Below each volcano plot, the complete list of genes per study are displayed with their log fold change (FC) and false discovery rate (FDR) values.

**Figure 3 genes-10-00759-f003:**
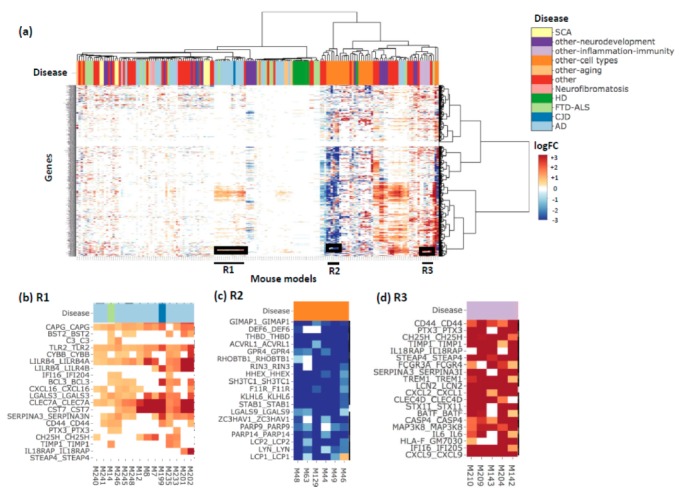
The yellow gene module heatmap. (**a**) The heatmap of genes from the yellow gene module from Zhang et al. [9] which is highly enriched with immune and microglial functions, annotated by disease. Each gene name is annotated as ‘Human-gene-symbol_Mouse-gene-symbol’. (**b**) A zoom in into the R1 boxed region from the heatmap where the majority of mouse studies are AD studies. (**c**) A zoom into the R2 boxed region where the studies were related to studying different cell types in mouse. (**d**) A zoom in into the R3 boxed region where all the studies are related to inflammation and immunity. Users can use the interactive heatmap available through the portal to zoom in to regions of interest.

**Figure 4 genes-10-00759-f004:**
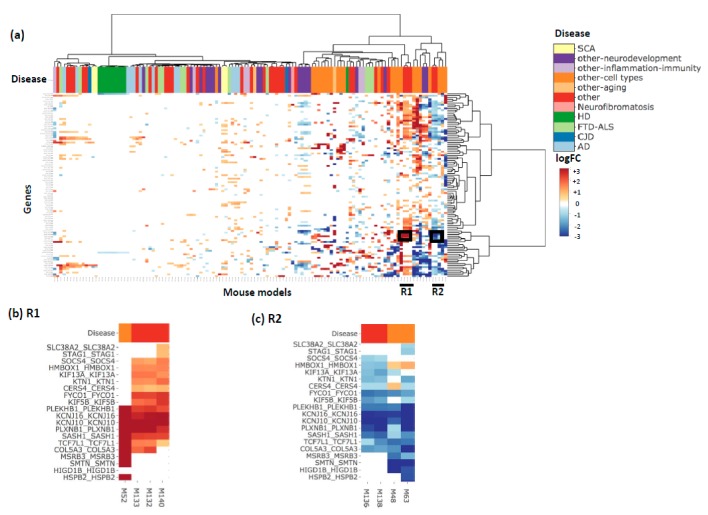
Module 109 (m109) heatmap. (**a**) The heatmap of genes from module m109 from Mostafavi et al. [10] which was significantly associated with cognitive decline, annotated by disease. Each gene name is annotated as ‘Human-gene-symbol_Mouse-gene-symbol’. (**b**) A zoom in into the R1 boxed region from the heatmap where M132, M133, M140 study the effect of ethanol on synaptic transcriptome and synaptic plasticity. (**c**) A zoom in into the R2 boxed region from the heatmap where M48 studies the transcriptome in neurons while M63 studies the transcriptome of dopaminergic neurons from the midbrain.

**Figure 5 genes-10-00759-f005:**
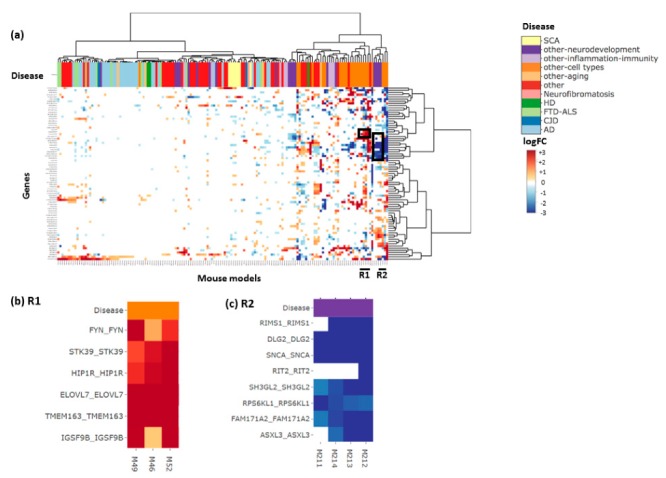
Examining nearest genes to top variants associated with PD. (**a**) The heatmap for nearest genes to single-nucleotide polymorphisms (SNPs) significantly associated with PD from a large PD genome-wide association study (GWAS) [13]. (**b**) A zoom-in into the R1 boxed region consisting of three DEGs (M49, M46, M52) studying cell types in the mouse. (**c**) A zoom-in into the R2 boxed region consisting of four DEGs (M211, M214, M213, M212) related to microglia development.

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
