# Peer review of "A Portal to Visualize Transcriptome Profiles in Mouse Models of Neurological Disorders"

_genes, 2019, doi:10.3390/genes10100759_

Round 1

Reviewer 1 Report

This is a very interesting and potentially very useful Web portal that has been put together in an imaginative and interesting way with a remarkable user interface.

What is however blatantly missing is reference to the original sources of the data referenced. The studies are listed as M69, M273 etc and there is a basic description of the genotype referred to and whether male or female mice and age and (rather vaguely) relevant disease but no way of finding the original manuscript or source. I am sorry if I am missing this somehow and it is in fact available; it is surely an essential element for the user to be able to refer back to the original source of data and how it was collected and in fact the text of the manuscript suggests that this might be what the user would want to do: "Faced with such a result, users might
investigate the original study so that they may better understand its experimental design to determine the causal mechanism." However, as far as I can see there is no way of doing this.

There are also other elements that are completely unannotated and unclear. What do the dendrograms on the heatmaps refer to and how are they attained? Possibly they are very useful but, as they are entirely unexplained, they just become noise on the sides of the plots.

Perhaps related to this, it is entirely unclear, except by guesswork, which of the annotated studies in fact come from different elements of the same study and which are from independent studies. The direct reference to the original studies would of course solve this omission.

Would it be possible to add a search for particular genes in the heatmaps so that if the user puts in a long list of genes, an individual gene can easily be found? It would also be useful if, under disease, the results could be restricted to one disease. This would greatly simplify the resulting heatmap and make it more easily searched. Moreover if annotating by disease grouping studies under the same disease together would be helpful. Similarly if under gender grouping those that were males and those females would help to bring out differences.

The results section of the paper illustrates a few examples of how the application could be used. They are successful in illustrating the basic outputs available which is the aim but it is rather unclear how the particular areas of the heatmap were chosen other than by a by-eye assessment of bits of the map, with many other areas showing similar patterns of differential expression.

In summary this is definitely a worthwhile approach and a potentially very useful resource but easy access to the sources of the data is essential and an improved description of all aspects of the plots would be helpful. This is true for people who are expert in analysing such data and even more so for those who are accessing and using the interface from other disciplines.

Reviewer 2 Report

In this manuscript, the authors present a description and representative output from an on-line query tool that accesses the extensive synapse.org housed AD-Cross-Species study (although apparently the data here accesses only mouse data and so the originating name is a bit of a misnomer). The objective of the work is to provide a web query tool that allows a user to type in/ upload/ paste a list of gene names and produce a heatmap, volcano plot(s), and a meta-study association table that will allow users to rapidly zero in on the appropriate model animal for a gene of interest. For example, if a user were interested in the role of Gene A in Disease X, and there were several animal models of Disease X that had been transcriptionally profiled, then an interested researcher could basically 'go shopping' for the 'right' animal model- that is, make sure that the animal model selected actually shows the gene changes in which the querying investigatory is interested. This is a good idea, and a lot of scientists could really use a tool like this to help in their research. However, some issues should be addressed.

Major-

Authors must include a caution on the consequences of gene-hacking (analogous to p-hacking) where researchers simply 'model shop' to find a gene of interest without regard to whether the gene expression change makes sense in the context of human transcriptional change, or whether the animal model, exclusive of that gene of interest, does in fact model the human condition.

Authors must include an evaluation of the similarity of each of these models to the human condition it is purported to study.

Authors must include a substantive and thorough discussion of other web portals and papers that summarize transcriptional profiling data from multiple studies, and explain the present work in the context of these other attempts.

Minor

Introduction- bottom line of first paragraph- authors have done a laudatory job of harmonizing data for comparison. However, whether the results are 'fairly' compared seems to be beyond the authors' scope

Overall- decision for the editor, but it is unclear if references to originating articles supporting public data submissions such as GEO should be cited in the bibliography

Web-based tool-

fonts for legend and for X axis are far too small (and are not zoomable). Therefore, major information that needs to be relayed to users is essentially not provided. This is also apparent in Fig. 2, 3, 4, 5- although it appears that some of the labeling in the manuscript for the zoom tool results is much more clear than that available in the current iteration of the web-tool.

Heat map should distinguish between values that are not being reported because of filtering strategies (e.g., a gene whose FDR is too high in a study), genes that are actually missing (e.g., not detectable in that study) vs genes that showed no change. Currently, all of these appear to be reported as if they were '0' changes. 

The Y axis of the heatmap is built from the uploaded list of genes. However, the x axis appears to be fixed based on the number of studies in the parent harmonized data set. This should be corrected so that the X axis is constrained to a user selected subset of all available studies

Study ID matrix should include the numerical values used to populate the heatmap, and shoud allow that information to be downloaded as a flat file.

 in Fig. 4, it looks like the comparisons are 'other-celltypes' and 'Other'- this should be clarified.

Round 2

Reviewer 2 Report

In response to my prior review, the authors have changed heatmap output (increased font size, provided options for display rows, included a download data button).

Their response to my request that they provide a review and discussion of other resources that summarize results across multiple studies was responded to with a data repository citations (GTEX, Brain-Seq), which is off-target.

In the original submission, the authors failed to cite prior publications whose data the present authors are using. In this revision, the authors' defend this position rather than correcting it by replying "we are including their data instead of their findings". The authors were largely unresponsive to other critiques regarding scope of work, and consequences of use, detracting from enthusiasm.
